# Empirical Analysis of IPv4 and IPv6 Networks through Dual-Stack Sites

**Kwun-Hung Li and Kin-Yeung Wong ***

School of Science and Technology, The Open University of Hong Kong, Hong Kong, China;
s1237552@ouhk.edu.hk
* Correspondence: akywong@ouhk.edu.hk

**Abstract:** IPv6 is the most recent version of the Internet Protocol (IP), which can solve the problem of IPv4 address exhaustion and allow the growth of the Internet (particularly in the era of the Internet of Things). IPv6 networks have been deployed for more than a decade, and the deployment is still growing every year. This empirical study was conducted from the perspective of end users to evaluate IPv6 and IPv4 performance by sending probing traffic to 1792 dual-stack sites around the world. Connectivity, packet loss, hop count, round-trip time (RTT), and throughput were used as performance metrics. The results show that, compared with IPv4, IPv6 has better connectivity, lower packet loss, and similar hop count. However, compared with IPv4, it has higher latency and lower throughput. We compared our results with previous studies conducted in 2004, 2007, and 2014 to investigate the improvement of IPv6 networks. The results of the past 16 years have shown that the connectivity of IPv6 has increased by 1–4%, and the IPv6 RTT (194.85 ms) has been greatly reduced, but it is still longer than IPv4 (163.72 ms). The throughput of IPv6 is still lower than that of IPv4.

**Keywords:** IPv6; IPv4; network performance; Internet; IoT

## 1. Introduction

Internet Protocol version 4 (IPv4) has been deployed as the core protocol of standards-based internetworking technology on the Internet and packet-switching networks in the past three decades. However, due to the rapid development of the Internet (particularly in the last decade), the allocation of IPv4 addresses has been exhausted. To overcome the address shortage problem, the Internet Engineering Task Force (IETF) created Internet Protocol version 6 (IPv6) in 1998. IPv6 provides sufficient address space for future use. It can also provide the feature of better routing performance. IPv6 has a great impact on the development of IoT devices [1,2] and senor networks [3–6].

Although IPv6 was launched over 20 years ago, the widespread deployment of IPv6 networks has only occurred in recent years. According to Google's statistics [7], from 2011 to 2020, the adoption of IPv6 increased dramatically. By September 2020, the deployment reached 30%, which is a great improvement compared to 2011. The statistics also show that the adoption was still increasing in 2020. The reason for the large-scale deployment of IPv6 in recent years is due to the exhaustion of IPv4 by the Internet Assigned Numbers Authority (IANA) [8] in 2011.

The performance of IPv4/IPv6 networks in 2004, 2007, and 2014 was empirically measured in References [9–11], respectively. These measurements were carried out by sending probing traffic from a dual-stack host (equipped with IPv4 and IPv6 protocol stacks) to dual-stack sites around the world. The results reveal that the deployment of IPv6 was still not extensive. Considering the recent widespread deployment of IPv6 networks, the goal of this study was to evaluate the performance of the latest IPv6 networks and compare it with that of IPv4. In this study, we followed the method used in [9–11], and we used connectivity, packet loss, hop count, round-trip time (RTT), and throughput as the performance metrics.

The results of our study provide IPv6 stakeholders with a picture of the current performance of IPv6 and IPv4 and their evolution over the past 16 years. This can enable stakeholders to make decisions about IPv6 deployment. For example, by strategically increasing bandwidth, adding more routers, or using faster IPv6 servers in some regions, more users can be attracted to use the network due to higher user satisfaction.

## 2. Related Work

Several studies have evaluated the performance of IPv4 and IPv6 from different perspectives. Jia et al. [12] tracked the IPv6 adoption by measuring core BGP networks and found that IPv6 networks have been slowly growing in recent years. They also discussed the geographical and topological inconsistencies in the deployment of IPv6. Moreover, it was shown that IPv6 had dynamic routing results similar to IPv4. While the above study focuses on measuring the core topology network, our study evaluates end-to-end traffic between end users and websites.

In addition to the performance of the core network, some studies [13–15] focused on the performance of IPv6 on 4G LTE networks. In [14], a dual-stack mobile device with LTE connectivity was used to measure the TCP connection establishment time to 40 websites worldwide. In [13], the round-trip latency performance between clients and servers, DNS lookup time, and web page load time for pages loaded over IPv6 and IPv4 networks were studied. The overall result in Reference [13] concluded that IPv6 is better than IPv4, which had the opposite result to Reference [14]. However, the mobile traffic is usually only in a dedicated area and does not involve global connections. The focus of this study is on global Internet traffic.

The performance of IPv6 and IPv4 under LAN and wireless LAN was considered in [16,17]. In [16], voice and video traffic on dual-stack networks were measured. Reference [17] analyzed the dual-stack mechanism on the wireless LAN as well as native IPv4 and IPv6 performance. This research evaluated the performance of voice, video, and FTP applications. Both papers show that IPv6 brings benefits to LAN and wireless LAN. Nonetheless, the experiments of these two studies were only conducted on the LAN, and the results may be affected by the throughput of local networks. Our research can avoid this problem since we sent probing traffic to dual-stack sites around the world.

In addition to previous studies, the use of dual-stack approaches in multimedia applications was evaluated. The research in [18] presented a comprehensive performance analysis and comparison of multimedia applications running on dual-stack networks. This study measured the performance of TCP and UDP protocols used in multimedia applications based on dual protocol stacks in the GNS3 simulator. The results show that IPv6 is more effective than IPv4 over TCP and UDP. Although the measurements in this study can reflect the performance of IPv4 and IPv6 over TCP and UDP, the results may have discrepancies because the tests were performed on simulators and virtual machines. Instead of simulators, our study used a real dual-stack computer to send probing traffic to the destination through physical networks.

From another perspective, how different transition and tunnelling methods affect the IPv6/IPv4 performance has been widely studied [19–24]. In [19], the performance of automatic tunnelling techniques, native IPv6, and IPv4 was evaluated. Dual-stack methods had better performance compared with tunnelling techniques. In addition, the native IPv6 had the highest throughput and the lowest RTT. Two other studies [20,21] used the GNS3 simulator and Wireshark to perform performance evaluation and analysis of dual-stack and manual tunnel transition techniques to investigate how data packets travelled through the network. Both studies evaluated static and dynamic dual-stack IPv4/IPv6 protocols and manual tunnelling techniques. References [22,23] explored and evaluated the performance of dual-stack, manual tunnel, 6to4 automatic tunnel, native IPv6, and IPv4 methods over real-time applications using the OPNET Modeler network simulator. The results showed that the dual-stack technique had better performance. Furthermore, native IPv6 had the worst performance in [22,23] since it is twice as long as the IPv4 header. Reference [24]

provided the background information of various transition mechanisms and analyzed IPv6 and IPv4 performance utilizing dual-stack and tunnelling techniques with GNS3. The result [24] reflected that dual-stack and tunnelling mechanisms had better performance than the native IPv6/IPv4 network. Nevertheless, different research results [22,23] showed that native IPv4 and IPv6 had better performance than dual-stack and other transition techniques. While both studies focus on transition and tunneling methods, the focus of this study is the dual-stack method.

From another perspective, two studies [10,11] conducted similar IPv6 and IPv4 performance analyses on dual-stack environments in 2007 and 2014. In [10], the dual-stack IPv6/IPv4 performance under various tunnel brokers was measured. The result showed that IPv6 had better connectivity and higher throughput but a lower hop count and higher latency. For the tunnel performance evaluation, FreeNet6 had the best performance result. Reference [11] sent probing traffic to dual-stack clients or servers in different regions to evaluate and compare the network performance of IPv6 and IPv4. The result revealed that the IPv6 packet loss rate and RTT were higher, and the number of hops and throughput was lower. Our study followed the methodology in [10,11] to evaluate the performance of today's IPv4 and IPv6 networks and measure the end-to-end experience.

This work was inspired by [10,11], which involved empirical performance studies of IPv6 and IPv4 under a dual-stack environment in 2007 and 2014, respectively. As reported in [10,11], the adoption ratios of IPv6 on 28 December 2007 and 13 December 2014 were 0.22% and 5.26%, respectively. We found that the ratio reached 32.26% on 11 OCT 2020. Considering such a high adoption rate, it is worthwhile to conduct similar research to study the performance of current IPv6 and IPv4 networks.

In this study, we adopted the measurement and comparison method in [11]. That is, we conducted empirical analysis of IPv4 and IPv6 performance by sending probing traffic from our testbed to 1792 dual-stack sites around the world. The performance metrics of our study are packet loss rate, round-trip time, hop count, and throughput. Then, we compared our results with the previous measurements in 2004 [9], 2007 [10], and 2014 [11] to observe how the performance changes as IPv6 adoption increases.

### 3. Methodology

*3.1. Measurement Setup*

This empirical measurement study was conducted in Hong Kong. Figure 1 shows our network testing environment. An end computer running dual stacks was connected to the broadband router. The router was connected, with a 100 Mbps link, to an ISP that provides IPv4 and IPv6 services.

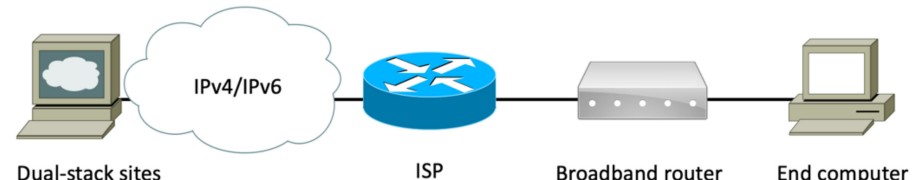

**Figure 1.** Network testing environment.

To evaluate the performance of IPv4 and IPv6, dual-stack sites from all over the world have to be identified beforehand. At first, we obtained the list of dual-stack sites or IPv6-enabled hosts from [10,25,26]. By consolidating the information from these three sources, the information of 5515 sites was obtained. The sites were then classified into regions and countries according to their top-level domain name. For the sites that could not recognize the geolocation based on the domain name (e.g., .com), the network utility tool of "nslookup" was used to determine the information. It queries the specified DNS server to retrieve the requested domain name or IP address, as shown in Figure 2. The website of iplocation.net was also used to identify the geolocation of sites, as illustrated in

Figure 3. However, the geolocation information of 125 sites could not be identified, and they were removed from the list. Thus, a total of 5390 dual-stack sites were considered in this study.

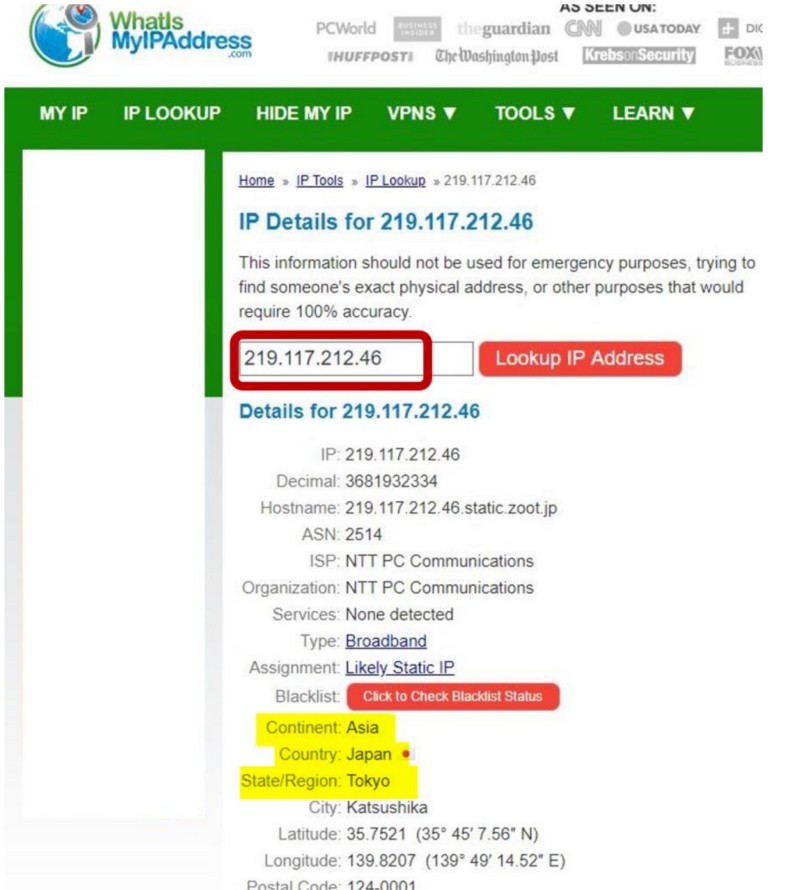

**Figure 2.** Example of executing nslookup on Command Prompt.

**Figure 3.** Example of checking the IP location on WhatIsMyIPAddress website, in which the continent of the IP address was indentified.

Though these sites claim to have dual-stack capability, not all of them can be accessed on both IPv4 and IPv6 simultaneously. The ping and ping6 network utilities were used to determine whether the sites could be reachable on both of them, as shown in Figure 4. The IPv4 and IPv6 reachability test results show that only 1792 sites are reachable via both IPv4 and IPv6 networks. The detailed ping test results in different regions are shown in Table 1. As a result, these 1792 sites were used in this study.

**Figure 4.** Example of IPv4/IPv6 ping test with batch script.

**Table 1.** The reachable dual-stack sites by region.

| Region | No. of Servers With Claimed or Possible Dual-Stack Capabilities | Pingable by both IPv4 and IPv6 |
|---|---|---|
| Africa | 59 | 9 |
| Asia | 1300 | 293 |
| Europe | 2662 | 1098 |
| North America | 951 | 273 |
| Central America | 55 | 6 |
| South America | 203 | 85 |
| Oceania | 160 | 28 |
| Overall | 5390 | 1792 |

*3.2. Measurement Methodology*

In this study, measurement traffic was sent to the 1792 dual-stack sites to evaluate the network performance in terms of (i) connectivity, (ii) round-trip-time (RTT), (iii) hop count, and (iv) throughput. The network utility tools used were ping [27], tracert [28], and wget [29].

Measurements were conducted from December 1st to 3rd, 5th to 8th, and 12th to 13th, 2020. To obtain fair results and eliminate the inaccuracy caused by network instability, the measurements were repeated three times (morning, afternoon, and evening), covering working days and public holidays.

*3.3. Throughput Test*

Throughput evaluations were carried by downloading files of different sizes (5 MB, 10 MB, 20 MB, 50 MB, 100 MB, 200 MB, 512 MB, and 1 GB) from the dual-stack sites. The wget [29] network utility tool was used to obtain the throughput results. The measurement was repeated 10 times a day.

## 4. Measurement and Results

### 4.1. Connectivity

The connectivity test was performed by the ping utility, which sent four packets to the sites and recorded the number of packets that could successfully reach the site.

Table 2 shows the results for the IPv4 and IPv6 packet loss rates on weekdays and weekends. It can be seen that the packet loss of IPv6 is 2.95% less than that of IPv4 on weekdays, and the loss of IPv6 is 0.35% less than that of IPv4 on weekends. Therefore, IPv6 sites have higher availability.

**Table 2.** Packet loss in connectivity test.

|  | IPv4 Packet Loss | IPv6 Packet Loss |
|---|---|---|
| Weekday | 3.95% | 1% |
| Weekend | 0.35% | 0% |

Table 3 shows the corresponding packet loss per region. On both weekdays and weekends, the IPv6 packet loss is lower than the IPv4 loss in all regions. During the weekend, IPv6 and IPv4 have similar results in all regions. However, during the weekdays, their results are quite different. For example, in Africa, the IPv4 loss is 5.56%, while the IPv6 loss is 0%. On the other hand, it is observed that among these regions, the networks in Europe are the most reliable as both IPv4 and IPv6 have meager loss rates (1.34% and 0%, respectively).

**Table 3.** Breakdown of packet loss by region.

| Region | IPv4 Weekday | IPv6 Weekday | IPv4 Weekend | IPv6 Weekend |
|---|---|---|---|---|
| Africa | 5.56% | 0% | 0% | 0% |
| Asia | 3.7% | 2.4% | 1.45% | 1.28% |
| Europe | 1.34% | 0.61% | 0.43% | 0.73% |
| North America | 4.3% | 2.04% | 0.27% | 0.18% |
| Central America | 4.17% | 0% | 0% | 0% |
| South America | 5% | 1.18% | 0.29% | 0.29% |
| Oceania | 3.57% | 0% | 0% | 0% |

### 4.2. Hop Count

A hop count measurement was performed to find the total number of routers that pass through to access the dual-stack sites. The hop count result was obtained by using the tracert [16] utility.

As can be seen in Table 4, IPv6 and IPv4 have a very similar average hop count, around 14.6. Figure 5 reveals the hop count distributions. It can be seen that IPv4 hop count resembles a right-skewed distribution, while IPv6 is comparable to a bell-shaped distribution. In addition, it shows that most IPv4 sites are centralized on a hop count of 12, while most IPv6 sites are concentrated on 14. Based on the results, in the context of hop count, it can be considered that the deployment of IPv6 has been developed to a level similar to that of IPv4. Table 5 shows the hop count divided by region. As can be seen, the hop counts of IPv4 and IPv6 are very similar for each of the regions.

**Table 4.** Average hop count for IPv4/IPv6.

|  | IPv4 | IPv6 |
|---|---|---|
| Average Hop Count | 14.66 | 14.61 |

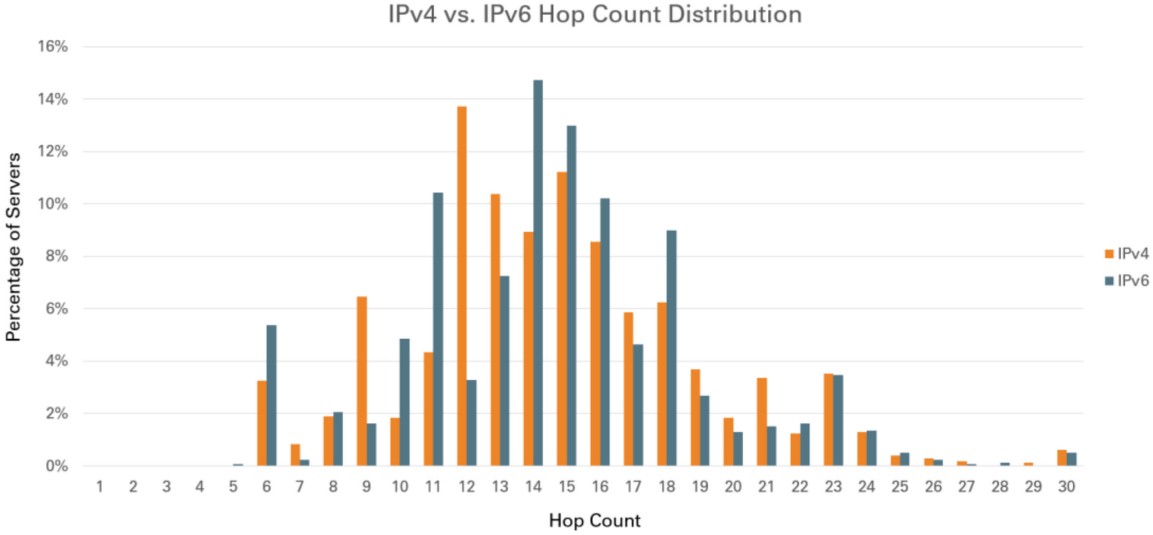

**Figure 5.** IPv4 vs. IPv6 hop count distribution.

The hop count distributions for Africa and Asia were studied and are shown in Figure 6. It was found that the distribution of hops in Africa is more even than that in Asia. The range of the distribution in Africa (from 5 to 16) is shorter than that in Asia (from 6 to 25). This implies that the deployment in Asia is more complex and extensive than that in Africa.

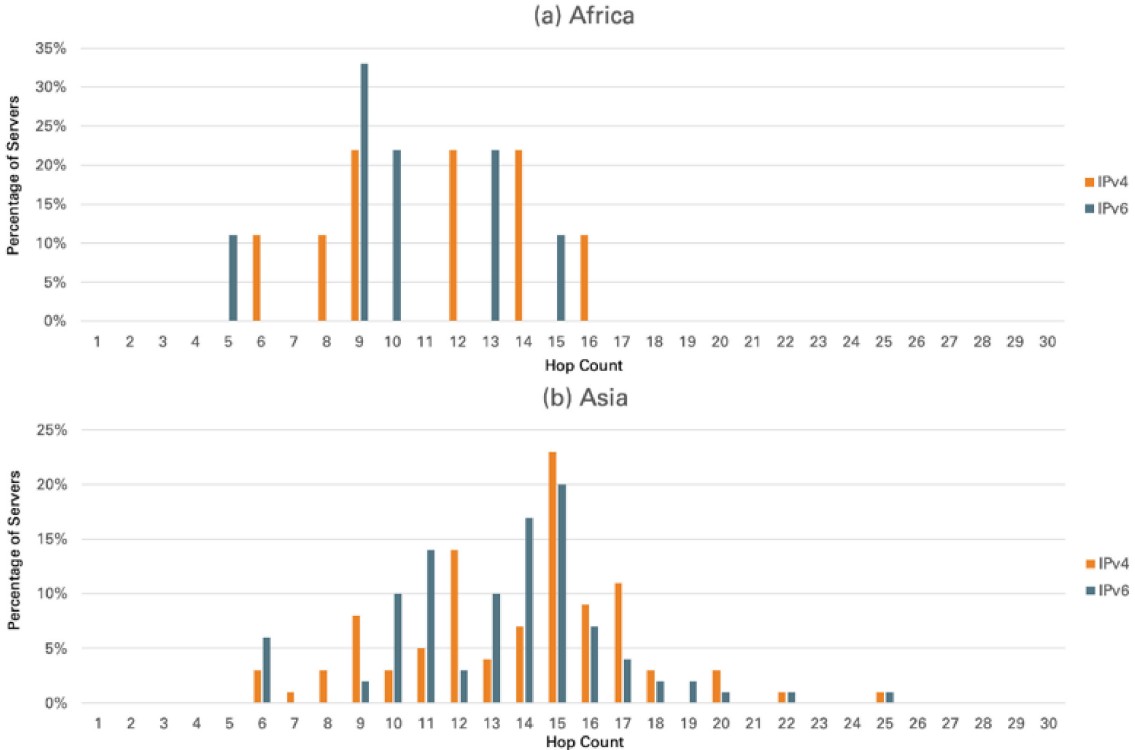

**Figure 6.** IPv4 vs. IPv6 hop count distributions for (**a**) Africa and (**b**) Asia. As can been seen, the distribution of Asia is wider than that of Africa.

The hop count can be regarded as the length of the path. It can also be used to access the status of IPv4/IPv6 deployment. This is because the increased number of hops is usually the result of widespread deployment. Note that a higher hop count does not mean higher delay; it is because widespread and new deployments usually involve faster routers

and more bandwidth, which could lead to lower round-trip time. The round-trip time in IPv4 and IPv6 networks is discussed in the next section.

**Table 5.** Breakdown of average hop count by region.

| Regions | IPv4 Hop Count | IPv6 Hop Count |
|---|---|---|
| Africa | 11.11 | 11.33 |
| Asia | 13.79 | 13.32 |
| Europe | 15.46 | 15.72 |
| North America | 12.33 | 11.71 |
| Central America | 9.67 | 10.67 |
| South America | 15.47 | 14.95 |
| Oceania | 14.43 | 13.71 |

*4.3. Round-Trip Time*

Round-trip time (RTT), also known as round-trip delay, is the time it takes to deliver a data packet to its destination, plus the time it takes from the acknowledgment packet sent from the destination to the source.

Table 6 shows the average RTT of IPv4 and IPv6 on weekdays and weekends. The RTT result shows that IPv6 is slightly higher than IPv4. The average RTT of IPv6 on weekdays is 194.97 ms, while IPv4 is 167.35 ms; on the weekends, IPv6 is 194.72 ms, while IPv4 is 160.08 ms. Table 6 also shows that there is only a slight difference in RTT between weekdays and weekends. For IPv4, there is about a 7 ms difference; for IPv6, the difference is about 0.25 ms. The distributions of IPv4 and IPv6 RTT are shown in Figure 7. It can be observed that there is not much difference between weekdays and weekends. It can also be seen that in the range of 300 to 400 ms (longest delay), the sites are mainly IPv6 sites.

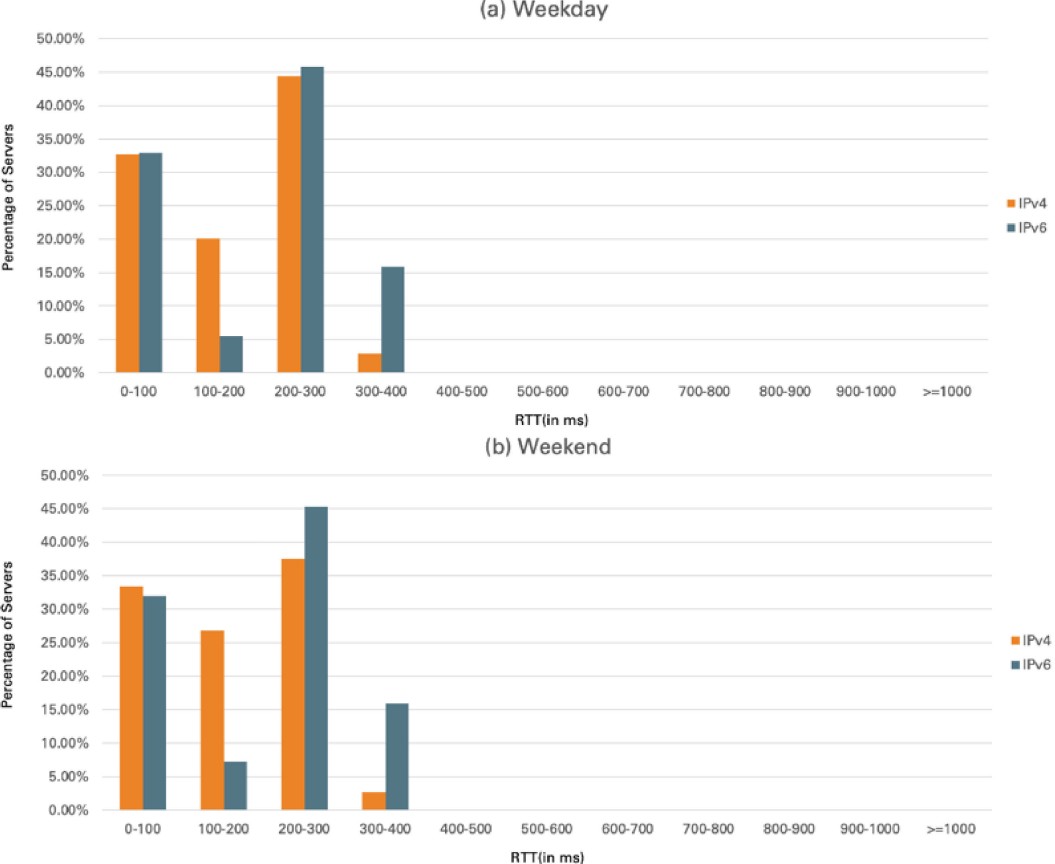

**Figure 7.** IPv4 vs. IPv6 RTT distribution.

**Table 6.** Round-trip time results.

|  | IPv4 Round-Trip Time | IPv6 Round-Trip Time |
|---|---|---|
| Weekday | 167.35 ms | 194.97 ms |
| Weekend | 160.08 ms | 194.72 ms |

Table 7 shows the RTT breakdown by region. As can be seen, the RTTs in Central America and Asia are the shortest among the regions, while those in Europe and South America are the longest. Figure 8 illustrates the RTTs for Central America and Europe for comparison (only weekdays are shown as the performances on weekends are very similar). It can be seen that some IPv6 sites have much longer delay. For example, in Central America, about 18% of IPv6 sites have delay in the range of 100 to 200 ms, while all IPv4 sites have less than 100 ms, as illustrated in Figure 8.

**Table 7.** Breakdown of RTT by region.

| Region | IPv4 Weekday | IPv6 Weekday | IPv4 Weekend | IPv6 Weekend |
|---|---|---|---|---|
| Africa | 114.41 ms | 107.78 ms | 100.56 ms | 106.78 ms |
| Asia | 62.71 ms | 60.04 ms | 59.05 ms | 59.13 ms |
| Europe | 211.15 ms | 257.06 ms | 202.17 ms | 256.90 ms |
| North America | 96.34 ms | 96.40 ms | 91.68 ms | 97.58 ms |
| Central America | 22.67 ms | 46.11 ms | 19.00 ms | 43.78 ms |
| South America | 221.61 ms | 224.24 ms | 215.45 ms | 222.10 ms |
| Oceania | 120.11 ms | 104.55 ms | 114.58 ms | 99.99 ms |

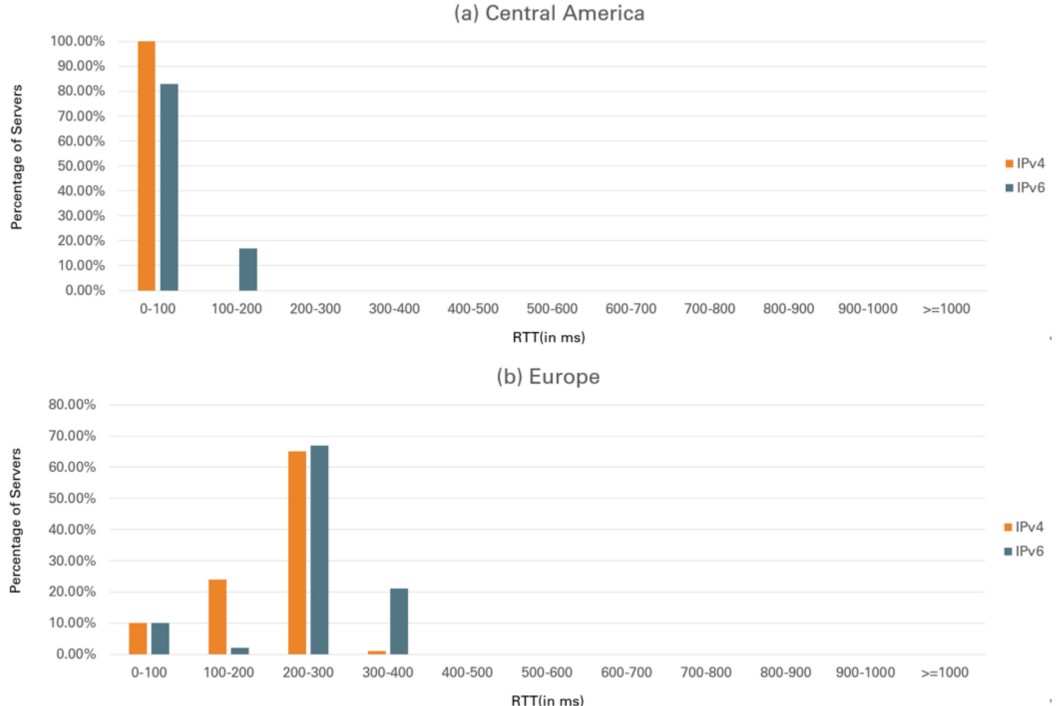

**Figure 8.** IPv4 vs. IPv6 RTT distribution on weekdays.

It can be seen from Section 4.2 and this section that the hop counts of IPv6 are less than those of IPv4, but the RTT of IPv6 is greater than that of IPv4. One reason is that more hops may mean more deployments. New deployments usually involve faster routers and more bandwidth, which indicates shorter round-trip times.

*4.4. Throughput Test*

A throughput study was conducted by uploading files to the dual-stack sites using IPv4 and IPv6 separately. The wget [18] utility was used to upload files. The throughput test was repeated 10 times on weekdays and weekends. In the throughput test, various file sizes were used: 5 MB, 10 MB, 20 MB, 50 MB, 100 MB, 200 MB, 512 MB, and 1 GB.

Table 8 shows the average throughputs of IPv4 and IPv6 using different file sizes on weekdays and weekends. As a result, it can be noticed that on weekdays and weekends, IPv6 throughput speeds of all different file sizes are slower than IPv4. For example, downloading a 50 MB file on the dual-stack server, on weekdays, the average throughput speed of IPv4 is 2238 KB/s, while that of IPv6 is 1504 KB/s. During the weekend, the average throughput speed of IPv4 is 5894 KB/s, while that of IPv6 is 4709 KB/s.

**Table 8.** Throughput test results.

| File Size | IPv4 Weekday | IPv6 Weekday | IPv4 Weekend | IPv6 Weekend |
|---|---|---|---|---|
| 5 MB | 780.70 KB/s | 645.20 KB/s | 1868.57 KB/s | 48.36 KB/s |
| 10 MB | 1081.60 KB/s | 792.80 KB/s | 3470.00 KB/s | 3284.00 KB/s |
| 20 MB | 1676.40 KB/s | 1082.00 KB/s | 3941.00 KB/s | 2580.50 KB/s |
| 50 MB | 2238.80 KB/s | 1504.30 KB/s | 5984.00 KB/s | 4709.00 KB/s |
| 100 MB | 2530.40 KB/s | 2470.80 KB/s | 6161.00 KB/s | 4982.00 KB/s |
| 200 MB | 4184.00 KB/s | 3941.00 KB/s | 7016.00 KB/s | 5854.00 KB/s |
| 512 MB | 4410.00 KB/s | 3968.00 KB/s | 7291.00 KB/s | 5976.00 KB/s |
| 1 GB | 7571.00 KB/s | 5569.00 KB/s | 6500.00 KB/s | 5052.00 KB/s |

Therefore, it can be observed that the throughput of IPv4 is still faster than that of IPv6, although IPv6 has already been widely deployed (in terms of hop count, as shown earlier).

## 5. Network Performance over the Past 16 Years

In the previous sections, the performance measurement results of today's Internet are discussed, and the test was performed in December 2020. In this section, the results of this study are compared with those obtained in 2004 [9], 2007 [10], and 2014 [11]. They can be compared because they use common performance metrics (connectivity, hop count, RTT for study [9–11], and throughput [11]), as shown in Table 9.

**Table 9.** Performance comparison with previous works.

| | 2004 [27] | 2007 [15] | 2014 [24] | 2020 |
|---|---|---|---|---|
| Number of dual-stack sites | 936 | 2014 | 2049 | 1792 |
| Connectivity IPv4 (Weekend) | 82.53% | 89.84% | 98.34% | 99.65% |
| Connectivity IPv6 (Weekend) | 73.98% | 96.57% | 97.35% | 100% |
| Hop Count (IPv4) | 17.5 | 19.81 | 17.05 | 14.66 |
| Hop Count (IPv6) | 8.7 | 15.22 | 14.3 | 14.61 |
| RTT (IPv4) | 281.84 ms | 272.78 ms | 252.75 ms | 163.72 ms |
| RTT (IPv6) | 409.80 ms | 403.36 ms | 418.06 ms | 194.85 ms |

As shown in the table, only 1792 dual-stack sites in 2020 can be identified for performance measurement. Compared with the study in 2004 [9], this is a significant increase in these 16 years. However, compared with the numbers in 2007 and 2014 [11], the number of dual-stack sites decreased by 222 and 257 sites, respectively.

Compared with the previous results, the connectivity of IPv6 and IPv4 has been improved in these 16 years. Compared with 2007 and 2014, connectivity of IPv6 has increased by 3.43% and 2.65%, respectively. In addition, we can observe that connectivity of IPv4 has increased by 9.81% and 1.31%, respectively. When compared to the measurement in 2004, both IPv6 and IPv4 networks have greatly improved. This shows that connectivity of IPv4 and IPv6 has increased by 26.02% and 17.12%, respectively.

Compared with previous works on IPv6 hops, the average hop count decreased by 0.92 hops from 2007 to 2014 and increased by 0.31 hops from 2014 to 2020. From 2004 to 2020, a total increase of 5.91 hops is observed. Although IPv6 has fewer hops than in 2007, from 2007 to 2020, the number of hops increased to more than 6 hops. For IPv4, we can see that from 2007 to 2014 and from 2014 to 2020, the number of hops decreased by about 3 hops. From 2007 to 2020, the IPv4 hop count continued to decrease and in 2020 it is less than in 2004. In short, the deployment of IPv4 (in terms of hop count) has decreased, the deployment of IPv6 was first reduced between 2007 and 2014, and the number of hops has grown slowly from 2014 to the present. It seems that an increasing number of networks have deployed IPv6.

By comparing RTT with these three studies, it can be found that both IPv4 and IPv6 now have lower latency. From 2014 to 2020, for IPv4, the average RTT was reduced by 89.03 ms, while for IPv6 it was reduced by 223.21 ms. From 2004 to 2020, IPv4 and IPv6 were reduced by 118.12 and 214.95 ms, respectively. However, the latency of IPv6 is still higher than the recent latency of IPv4, as observed for the RTT results in 2004, 2007, and 2014. In short, the RTT of IPv6 has been greatly enhanced, but it still needs to be improved. This is a positive sign of better deployment of IPv6 in recent years.

The throughput results obtained in this study are similar those in 2014 [11]. IPv6 throughput is still lower than IPv4. In the past few years, the speed of the IPv4 network has been similar.

Please note that our performance measurements were conducted in Hong Kong, and the overall results may not be the same if the originating location is different. We also only followed the IPv4/IPv6 dual-stack approach (which only includes around 35% of the total hosts in the IP world).

Nevertheless, the empirical measurements by [9–11] and the studies in 2004, 2007, 2014, and 2020 were all conducted in Hong Kong. Therefore, this study (in Hong Kong) could provide meaningful insights from a consistent perspective into the development of IPv4 and IPv6 in the past 16 years.

## 6. Conclusions

The first contribution of this study was to conduct of an empirical analysis of the performance of IPv6 and IPv4 networks. Connectivity, packet loss, hop count, round-trip time (RTT), and throughput were used as the performance metrics. The results show that IPv6 has a lower packet loss rate. The results also show that the average hop count of the IPv6 network is very similar to that of IPv4, which implies that IPv6 has been deployed to a level similar to that of IPv4. IPv6 has a higher RTT and lower throughput than IPv4. Another contribution of this study is that the results were compared with previous studies in 2004, 2007, and 2014. This comparison is significant because it can determine the development of IPv6 networks in the last decade. The comparison shows that in the last 16 years, the connectivity of IPv6 has improved by 1–4%, and the IPv6 RTT has been greatly reduced, but it is still longer than that of IPv4. The throughput of IPv6 is still lower than that of IPv4, and there is no noticeable improvement since 2014.

Our research results provide IPv6 stakeholders with a picture of the current performance of IPv6 and IPv4 and their evolution over the past 16 years. This can enable stakeholders to make decisions about the strategies of IPv6 deployment.

Considering the trend of deploying IPv6 networks, we plan to perform similar research in two to three years. We may use a virtual private network (VPN) to connect to other computers in different regions (e.g., Europe and North America) in order to provide more comprehensive measurement results.

**Author Contributions:** Conceptualization, K.-Y.W.; methodology, K.-H.L. and K.-Y.W.; writing—original draft preparation, K.-H.L.; writing—review and editing, K.-Y.W.; supervision, K.-Y.W.; project administration, K.-Y.W.; funding acquisition, K.-Y.W. All authors have read and agreed to the published version of the manuscript.

**Funding:** The work described in this paper was fully supported by the Open University of Hong Kong Research Grant (School Project R5089 [2020/21 S&T]).

**Institutional Review Board Statement:** Not Applicable.

**Informed Consent Statement:** Not Applicable.

**Data Availability Statement:** In this section, please provide details regarding where data supporting reported results can be found, including links to publicly archived datasets analyzed or generated during the study. Please refer to suggested Data Availability Statements in section "MDPI Research Data Policies" at https://www.mdpi.com/ethics (accessed on 12 September 2020.). You might choose to exclude this statement if the study did not report any data.

**Conflicts of Interest:** The authors declare no conflict of interest.

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
