# Peer review of "Empirical Analysis of IPv4 and IPv6 Networks through Dual-Stack Sites"

_information, doi:10.3390/info12060246_

Round 1

Reviewer 1 Report

The reviewer has the following comments.

  1. This empirical measurement study was conducted at a site of Hong Kong.
  2. Figure 1 shows the HKT modem instead of the broadband router mentioned in the paper. The authors should introduce the information about the router and the backhaul link to the Internet since these may affect the results.
  3. Using DNS query can only identify that the DNS has recorded the IPv4 and IPv6 addresses of a host. That cannot identify that the host has both IPv4 and IPv6.
  4. The IPv4 an IPv6 address of the hosts on the DNS server are added by the network administrator, but the user may not set these IP addresses for the host.
  5. The "Whois Database" query only identifies the owner of the IP addresses.
  6. The authors should identify the site list of the dual-stack sites by region. The reviewer cannot verify that whether the list is complete or good enough for this study.
  7. The measurement only presents the performance indexes between the site and the testbed.
  8. The results should be further explained. Take hop count, for example, the authors should explain why the hop count is the same or different.
  9. The hop counts of IPv6 are less than that of IPv4, but the RTT of IPv6 is more than IPv4. The authors can explain these results based on the network topology information.

Reviewer 2 Report

Which specific method of comparison has been adopted for this study is not mentioned in the abstract, even no comparative result statistics are given.

The authors have tangled the introduction and the related work section each other. For the introduction of the proposal, it is crucial to highlight the contribution which has not seen anywhere in the whole section. Further, the subject matter is not presented in a comprehensive manner and English writing and grammar are very weak; also the selection of words is not easily understandable. In addition, the structure of the paper is weak; in particular, the introduction contains the description of the previous studies that should come later in the latter section like Related work. It is not clear what to expect in the paper, what the flow/structure is (where to find what), etc.

While considering the related work section, the authors have failed to highlight the shortcoming of the previous work. A similar work regarding IPV6 is available with the same context but it is imperative to discuss the limitation. If the previous work is free of challenges and there is no issue then what motivated to authors to proposed this study? It inculcates that authors should revised the literature review and critically highlight the problems in the previous study and compare the proposed solution and tells how does the proposed solution is best fitted.

After reading through the work, I'm not convinced that any model was developed. It is only an application of an existing model to analyze some data. So there may be the need to adjust the substantial findings to reflect what was actually done. And if not, the author needs to emphasize in the methodology how the model was developed.

Similarly, in the Methodology section; The article missed presenting the research novelty and proposed framework. In fig2 and 3, the authors are showing the result of nslookup command! If I'm not wrong, what relation they are going to build with the proposal?

I strongly recommend that the Flow chart or the algorithm of the proposed work should be incorporated and it is also advised to critically present a technical method using an algorithm, mathematical computation for the proposed methodology with the help of a flowchart.

The authors have tried to cover the result discussion with all efforts but without specifying the foundation.

The machines are operating on virtual addresses/ IP addresses therefore, It is crucial that "device coordination is an important parameter", however, it is obvious that such coordination may essentially need a synchronization aspect to implement. How it has been taken care of in the present paper? If so then whether overhead has been considered or not?

From fig5, the comparison of hop count is being illustrated. It is important to discuss how does the hop count is being calculated? Please mention the name of that method and the related description with proper citation and then analysis your result in figure 5 and also mention what measures will be taken if the proposed analysis going to fail.

The RTT breakdown is showing in fig 8, this evaluation is somewhat confusing or incorrect in the relevant Table. Please revise your discussion.

Please points out some insufficiency and limitation that needs further improvements in the conclusion. Moreover, formats of reference lists lack consistency. The conclusion should be rewritten to clarify the contribution and future research direction should be given in this section.

There are numerous places in the text with English grammatical errors. The authors should be full checking for grammar and mistakes to meet the quality of the Journal.

The current references are acceptable but not enough therefore, it is highly recommended to add more relevant citations in order to improve the value of this study. Some references are recommended in this regard

https://dx.doi.org/10.26555/jiteki.v6i1.17105

https://dx.doi.org/10.17352/tcsit.000012

https://doi.org/10.26634/jwcn.8.3.17310

Reviewer 3 Report

The paper has a non-sophisticated approach and the descriptions of the methodology, measurements and results are straightforward. The readers are attracted from the beginning to find if the performance of these IPv4/IPv6 dual-stack sites evolved since previous tests started almost two decades ago. However the easy to understand presentation (an advantage for the readers !) seems to generate on the other hand legitimate questions about the relevance of the results. For instance if we conduct the experiment by placing PC1 and PC2 (see Figure 1) on a different continent (e.g. Europe), do we expect to get the same conclusions? I am referring here mainly to the absolute values of RTT distribution obtained in Table 6, Figure 7 and 8.  I presume the distribution does depend on the global topology.  Other suggestions to improve the paper:

  • typo at R223
  • Do not start a phrase with "And.." (R206)
  • the quality of the figures (the resolution is not good enough)
  • I do not think you are entitled at the end of the paper to conclude that "...IPv6 has not improved since 2014".

Overall you should better explain how do you exploit the experimental results obtained from this paper. 

Round 2

Reviewer 2 Report

The authors have responded very well but at some point I'm still in confusion and neither convinced with the logic provided by the authors.  I found some superfluous citations which should be avoided like [21], [24], [26] and [27]. I suggest replacing all these citations with the below-recommended references.

Shahzad Ashraf, Durr Muhammad, Zeeshan Aslam (2020). Analyzing challenging aspects of IPv6 over IPv4. Jurnal Ilmiah Teknik Elektro Komputer Dan Informatika, 6(1), PP.54-67. doi: https://dx.doi.org/10.26555/jiteki.v6i1.17105

Ashraf, S (2019). Culminate Coverage for Sensor Network through Bodacious-Instance Mechanism. i-manager’s Journal on Wireless Communication Networks , 8(3), 1-9, doi: https://doi.org/10.26634/jwcn.8.3.17310

S. Ashraf, M. Gao, Z. Chen, H. Naeem, A. Ahmad and T. Ahmed, "Underwater Pragmatic Routing Approach Through Packet Reverberation Mechanism," in IEEE Access, vol. 8, pp. 163091-163114, 2020, doi: https://doi.org/10.1109/ACCESS.2020.3022565

Ashraf, S., Ahmed, T., Aslam, Z., Muhammad, D., Yahya, A., Shuaeeb, M. (2020). Depuration‎ based Efficient Coverage Mechanism for ‎Wireless Sensor Network. Journal of Electrical and Computer Engineering Innovations (JECEI), 8(2), 145-160. doi: 10.22061/jecei.2020.6874.344

Author Response

[26] is removed.
[21][24][27] will be kept because [21] talks about dual-stack on multimedia applications and [24][27] are core references in our result section.

The four proposed references have been added as [26] [29][30][31].

Reviewer 3 Report

I appreciate your effort to address the comments. However I am still not convinced by your explanation "If the measurement is originated in another region, say, in Europe, we believe that the value of particular performance metric (not matric!!!) will be different (e.g., the hope count between Europe to Africa is different from that between Hong Kong to Africa). However, the overall performance comparison between IPv4 and IPv6 will be similar".

Because the weakest link counts in measurements on the Internet, the results are fully dependent on topology between the source and the destinations. According to https://blog.apnic.net/2021/02/08/ipv6-in-2020/ (Figure 9) the deployment of IPv6 in January 2021 shows the bottlenecks in the performance test (originating from Hong Kong as in your paper).

I understand that you are following an approach launched many years ago (and this is a merit of your work). Thus, to compare the results you need to launch the tests from the same location (and not from India or US where the IPv6 deployment seems to be better). My suggestion is to add a note mentioning that the overall results may not be the same if the originating location is different. Also, a disclaimer that you followed just IPv4/IPv6 dual-stack approach (which only includes around 35% of the total hosts in the IP world).

Some minor typos in the newly introduced text:

R48 "metrics" instead of "matrics"

R160 "nslookup" instead of "Nslookup" (for the operating systems which are case-sensitive...).

Author Response

We appreciate the comments very much.
1. In the revision, the explanation that you are not convinced with will be removed.

2. In section 5, I added a note mentioning that the overall results may not be the same if the originating location is different. And included a disclaimer that you followed just IPv4/IPv6 dual-stack approach (which only includes around 35% of the total hosts in the IP world).

3. The typos have been corrected.